# Factors Associated with Spontaneous Clearance of Recently Acquired Hepatitis C Virus among HIV-Positive Men in Brazil

**DOI:** 10.3390/v15020314

**Published:** 2023-01-23

**Authors:** Rosario Quiroga Ferrufino, Camila Rodrigues, Gerusa Maria Figueiredo, Daniel Gleison, Silvia Yapura, Maria Laura Mariano de Matos, Steven S. Witkin, Maria Cássia Mendes-Correa

**Affiliations:** 1Departamento de Molestias Infecciosas e Parasitarias, Aculdade de Medicina, Universidade de São Paulo, Av. Dr. Enéas Carvalho de Aguiar, n. 470, São Paulo 05403-000, Brazil; 2Hospital das Clínicas, Faculdade de Medicina, Universidade de Sao Paulo, São Paulo 05403-010, Brazil; 3Instituto de Medicina Tropical, Faculdade de Medicina, Universidade de São Paulo, São Paulo 05403-000, Brazil; 4Departamento de Medicina Preventiva, Faculdade de Medicina, Universidade de São Paulo, São Paulo 05403-000, Brazil; 5Department of Obstetrics and Gynecology, Weill Cornell Medicine, New York, NY 10065, USA

**Keywords:** HIV/coinfection, recently acquired hepatitis C infection, HIV, spontaneous clearance, immunity, acute HCV

## Abstract

Introduction: The objective of the present study was to describe the clinical and epidemiological aspects of recently acquired hepatitis C virus (HCV) infection and the frequency of its spontaneous clearance in a people living with the human immunodeficiency virus (PLWH) cohort. Methods: We reviewed the medical records from all PLWH at the human immunodeficiency virus (HIV) outpatient reference clinic affiliated with the University of São Paulo, Brazil, and identified, by immunoassays and RNA-PCR individuals who acquired HCV infection between January 2015 and December 2017. The factors associated with subsequent spontaneous clearance of the infection in this group were identified and analyzed. Results: Among 3143 PLWH individuals, 362 (11.5%) were coinfected with HCV. Forty-eight (13.2%) of these subjects first became HCV-positive between January 2015 and December 2017. Spontaneous HCV clearance was documented in 23 individuals (47.9%). The majority of this latter group were male (83.3%), and the median age was 31 years (23–39). The main risk group for HCV acquisition was men who had sex with men (MSM) (89.5%). In a multivariate analysis, only an elevated CD4+ T lymphocyte count at the time of seroconversion was found to be associated with subsequent HCV clearance (*p* = 0.025). Conclusions: In HIV-infected individuals in Sao Paulo, Brazil, most cases of recent HCV transmission were by sexual exposure. In PLWH, particularly in MSM, the individual’s CD4+ T lymphocyte count is a determinant of whether an acquired HCV infection will be prolonged or will spontaneously clear.

## 1. Introduction

There are an estimated 58 million people with chronic hepatitis C virus (HCV) infection globally, resulting in substantial morbidity and mortality [1]. The major routes of HCV transmission are currently well described. There is, however, great global variation in the rates of different transmission routes [2]. Transfusion of blood products before the availability of effective HCV testing of blood donations had been the major route of transmission in most countries worldwide [3]. Before the onset of donor HCV screening, the rate of post-transfusion hepatitis, now known to be attributable to HCV, reflected the background prevalence of the infection in the blood donor population [2]. Among intravenous drug users (IVDU) the prevalence of HCV infection ranges from 45% to >90%, and annual incidence rates are 6–40% [4]. The risk of mother-to-infant transmission has been estimated at around 6% from HCV monoinfected mothers and twice as high from HCV–human immunodeficiency virus (HIV)-coinfected mothers [5]. For needlestick injuries pooled estimates of the risk of HCV transmission from an infected source are 0.5–1.9% [2]. Amongst heterosexual couples, HCV transmission through sexual activity is rare [6,7,8]. 

Recently, however, several studies have documented an increased risk of transmission amongst individuals with multiple sexual partners, established sexually transmitted disease, and HIV infection [9,10]. Outbreaks of recently acquired HCV infections in men who have sex with men (MSM), particularly in those living with HIV (PLWH), have been observed globally. Most of these outbreaks have been described in Europe, Australia, and the United States [11,12,13,14,15]. However, in Brazil and other countries in Latin America, there is a paucity of data on the mode of transmission of HCV infection in PLWH.

The proportion of patients who spontaneously clear HCV infection varies between studies, but it is believed to range from 20–50% and 5–20% among monoinfected and HIV-coinfected individuals, respectively. Other factors have also been nominated as predictors of HCV clearance, such as polymorphisms in the interleukin-28 (IL-28) gene, female gender, and symptomatic disease [16].

Characterizing epidemiologic aspects of HCV infection among a vulnerable population such as PLWH and analyzing rates of spontaneous viral clearance is essential for assessing the long-term burden of disease and to determine the natural course of HCV among this high-risk population. 

The objective of the present study was to describe the clinical and epidemiological characteristics of recently acquired HCV infection and the frequency of its spontaneous clearance in PLWH attending a reference center in Brazil.

## 2. Materials and Methods

### 2.1. Study Design and Population 

This was a retrospective, cross-sectional study performed at the AIDS Outpatient Clinic, a reference clinic affiliated with São Paulo University Medical School in Brazil. The study enrolled all patients (n = 3143) in follow-ups at the institution from January 2015 to December 2017.

Adult clinic attendees living with HIV undergo systematic HCV serological screening every 6–12 months. We initially reviewed medical records from all patients to identify HCV-infected individuals. HCV infection was diagnosed by detection of anti-HCV antibody or anti-HCV antibody and HCV-RNA. 

Among them, we searched for individuals who had recently acquired HCV infection.

Recent HCV infection was defined as being positive for anti-HCV antibody or anti-HCV antibody and HCV-RNA beginning no earlier than January 2015, with negative anti-HCV antibody tests in the previous 6–12 months. 

The seroconversion date was defined as the first positive anti-HCV antibody test.

Among this latter group we further identified individuals who subsequently achieved spontaneous HCV clearance. Spontaneous clearance was defined as being HCV RNA-negative in at least two consecutive serum samples, collected during a period of a minimum of 1 year of follow-up after the estimated seroconversion date, without previous treatment for HCV [17]. 

### 2.2. Laboratory Tests 

Initial detection of HCV was performed by using an ARCHITECT Anti-HCV high throughput chemiluminescent microparticle immunoassay (Abbott Laboratories, Chicago, IL, USA). 

For HCV-RNA detection we performed real-time polymerase chain reaction (PCR) with a detection limit of 12 IU/mL (Abbott Molecular, Des Plaines, IL, USA).

For HCV genotype, samples were tested using the Abbott Real Time HCV genotype II assay.

HIV viral load was assessed by using a Real-time HIV-1 Viral Load Assay by Roche, TaqMan HIV-1, version 2 (COBAS v2.0), or Abbott m2000 Real-time HIV-1 (m2000rt) with a detection limit of 20 to 40 copies/mL.

### 2.3. HCV Spontaneous Clearance

For the analysis of factors associated with spontaneous clearance of HCV, we included age, gender, and multiple risk factors for HCV transmission such as MSM, IVDU and blood transfusion, median CD4+ T lymphocyte count level, HIV RNA level, median maximum alanine transaminase (ALT) and aspartate transaminase (AST) level (U/L), HCV genotype, and number of comorbidities and opportunistic infections prior to HCV diagnosis. The time between HIV and HCV diagnosis and history of previous liver disease at the time HCV diagnosis were obtained from medical records. Data on clinical symptoms and HCV treatment, as well as on hepatitis B virus (HBV) and syphilis status, were also analyzed. Information on ALT/AST level, HIV viral load, and CD4+ T lymphocyte counts was extracted from the data during the six months prior to HCV diagnosis. 

### 2.4. HCV Treatment

HCV treatment was performed with direct acting agents (DAAs), according to the Brazilian Ministry of Health Hepatitis C Guidelines, which recommend HCV therapy for all HIV-coinfected individuals with acute or chronic HCV infection. All individuals included in this study received 400 mg sofosbuvir/60 mg daclatasvir, in one tablet, once daily for 12 weeks. 

Sustained virologic response (SVR) was defined as the absence of detectable HCV RNA in serum at the end of treatment and six months later.

### 2.5. Statistics 

Qualitative data were provided as absolute and relative frequencies, and quantitative data were described as summary measures for all patients. Values were expressed as a mean (range) or median (IQR). HCV spontaneous clearance was expressed according to each qualitative variable.

Continuous variables were compared using the student t-test or the Mann–Whitney U-test when applicable. Fisher’s exact test was applied to evaluate differences between categorical variables.

Bivariate logistic regression was used for each variable to estimate an odds ratio (OR) with 95% confidence intervals for spontaneous clearance of HCV. All statistically significant variables in the bivariate analysis were used in a multivariate logistic regression analysis to identify independent variables that influenced spontaneous clearance of HCV. The analyzes were performed using the IBM-SPSS for Windows version 22.0 software and tabulated using the Microsoft Excel 2010 version 14.0 software. All tests were performed at a 5% significance level.

## 3. Results

Among 3143 HIV-infected individuals in our cohort, 362 (11.5%) were identified as being HCV-infected, by being either positive for anti-HCV antibody or positive for anti-HCV antibody and HCV-RNA. 

Forty-eight (13.2 %) of these individuals became HCV-seropositive between January 2015 and December 2017, fulfilling the inclusion criterion. Forty (83.3%) were male, with a median age of 31 years. 

The major risk group for HCV seroconversion was MSM (89.5%). All subjects denied IVDU or being the recipient of a blood transfusion. 

The time of follow-up before HCV seroconversion was documented and varied from 13 months to 16 years. Each of the 48 individuals was undergoing antiretroviral therapy. HIV RNA was undetectable in 85.4% of them, and their median CD4+ T lymphocyte count was 753 (1050-456) cells/µL. The specific HCV genotype was available for 22 (45%) of the individuals. Genotypes 1, 4, and 2 were identified in thirteen (59%), seven (31.8%), and two (9%) individuals, respectively.

Only 1 individual was positive for HBV surface antigen; 29 were positive for antibodies to HBV core antigen, while 23 were positive for anti-HBV surface antibodies. Twenty-two were treated for syphilis during the six months prior to their HCV diagnosis. Baseline characteristics are summarized in Table 1. 

Most patients were asymptomatic at the time of their HCV diagnosis, and only seven reported mild-to-moderate symptoms related to their recently acquired HCV infection (positive HCV PCR test with a negative anti-HCV test in the previous 6–12 months) such as diarrhea, fever, abdominal pain, asthenia, myalgia, icterus, nausea, and vomiting.

Only one patient developed severe symptoms and was hospitalized for one week. Among the patients, five were treated with direct-acting antiviral (DAA) medications and reached an SVR. Two exhibited spontaneous HCV clearance. Among the symptomatic patients, the median elevations of AST and ALT were 2.98 ± 6.48 and 4.4 ± 7.76 above the upper limit of normality, respectively.

Spontaneous HCV clearance occurred in 23 individuals (47.9%). Twenty-five individuals were repeatedly PCR-positive when tested at least twice over a minimum follow-up period of 12 months and were considered to have not spontaneously cleared HCV. Among the latter subjects, 21 received HCV treatment with DAAs, according to the Brazilian Ministry of Health Hepatitis C Guidelines. All 21 individuals received 400 mg sofosbuvir/60 mg daclatasvir, in one tablet, once daily for 12 weeks. Four patients were lost to follow-up. All 21 DAA-treated patients did not receive medication immediately after their HCV diagnosis. Due to DAA prescription restrictions in Brazil, the time interval between prescription and treatment was usually longer than six months. An SVR was observed in all 21 patients. 

In a univariate analysis, spontaneous HCV clearance was positively associated with female gender (*p* = 0.020), high CD4+ T lymphocyte count (*p* = 0.006), and the absence of ALT or AST elevation levels six months prior to HCV clearance (*p* < 0.001) (Table 2). 

In a multivariate analysis, only a high CD4+ T lymphocyte count was associated with HCV clearance (*p* = 0.025) (Table 3). For each increase of 100 cells in CD4+ T lymphocyte count, there was a 4.5% increase in the chance of spontaneous HCV clearance.

## 4. Discussion

To the best of our knowledge, this study is the first epidemiological examination of recently acquired HCV infection among a large sample of PLWH in Brazil. HCV infection was predominantly acquired among MSM, and a high rate of spontaneous clearance of HCV infection was identified.

The main mechanism of HCV acquisition among PLWH in our study was by sexual exposure. This mode of HCV transmission has been reported in several investigations worldwide from the 2000s onwards, predominantly in HIV-positive MSM [9,10,11,12,13,14,15]. Our findings are in accordance with these observations.

In Brazil, in the past, blood transfusion and IVDU were the main risk factors associated with HCV acquisition in the general population and among HCV–HIV-coinfected individuals [18]. In a recent systematic review in Brazil, Tengan et al. estimated an HCV–HIV coinfection prevalence of 18.9% [19]. Mandatory screening for HCV in blood banks beginning in 1993 and changes in Brazil’s illicit drug market in the past decade have markedly decreased national HCV transmission by these modes [20,21]. Nowadays, IVDU is relatively rare in Brazil, reportedly occurring in only 0.4% of persons 12 to 65 years of age during their lifetime [22]. According to the Third National Survey of Alcohol and Drugs by Brazil’s National Institute for Public Policy Research on Alcohol and Other Drugs (2019), marijuana is the most consumed illicit drug in Brazil followed by snorted or smoked cocaine (crack) among the general population that is 12 to 65 years old [22]. Although uncommon, the use of injectable cocaine has been recently documented in a few studies in the Brazilian Amazon region among people who use illicit drugs [23,24].

Our data regarding the modes of HCV transmission among HIV–HCV-coinfected individuals probably reflect changes in the safety of blood transfusion as well as in the structure and modes of distribution of illicit drugs in this country over the past several years. 

It is plausible to also suppose that in the past a constant level of HCV sexual transmission may have always occurred. However, the high levels of IVDU-related HCV transmission might have masked HCV sexual transmission. These abovementioned factors and specific sexual practices of MSM likely contributed to the proportion of HCV cases due to sexual transmission in our cohort. High-risk practices among a subpopulation of these individuals, such as anal intercourse with a high number of sexual partners and recreational drug use to enhance pleasure, along with the widespread use of smartphone apps and online “hooking-up” sites to further increase the availability of casual sexual encounters, could contribute to this situation [12,13,14,15]. 

Most patients in our cohort were asymptomatic at the time of their HCV diagnosis, and only seven reported mild-to-moderate symptoms related to their recently acquired HCV infection. Our findings reinforce the fact that due to the asymptomatic nature of recently acquired HCV infection, it can be difficult to diagnose in the early stage of infection. Physicians should routinely screen at-risk individuals and investigate abnormal liver function tests.

A high rate of spontaneous HCV clearance, in 47.9% of 48 individuals who had recently acquired HCV infection, was observed in our cohort. This rate is high when compared with previous studies that showed an 11% to 49% spontaneous HCV clearance rate in different populations [25,26,27,28,29,30]. 

A recent meta-analysis has verified that the proportion of spontaneous HCV clearance varies over time, combined with an assessment of demographic, clinical, and behavioral determinants. The study revealed that the rate of spontaneous viral clearance increased over time, occurring in 19.8%, 27.9%, 36.1%, and 37.1% of individuals within 3, 6, 12, and 24 months after infection, respectively [16]. Other studies have observed lower rates of virus clearance. Most reported clearance rates of approximately 25% without DAA treatment [25]. Differences among these studies may be related to the heterogeneous groups of patients involved and to different times of observation after infection. Variables not associated with HCV clearance in these studies were male sex, presence of an asymptomatic infection, black race, older adults (age ≥ 45 years), HIV coinfection, absence of HBV coinfection, non-genotype 1 HCV infection, nonaboriginal groups, and a history of IVDU [16,25,31,32,33,34,35]. Conversely, polymorphisms in the interleukin-28 (*IL28B*) gene have been recognized as the strongest genetic factor associated with HCV clearance [16,25,32,33,34,35].

In our cohort, most individuals were MSM. Sexual transmission has been associated with high rates of spontaneous HCV clearance. The reasons for this are not clear and remain controversial. It has been hypothesized that a smaller viral inoculum (compared to acquisition of the virus intravenously) might account for the ability of the immune system to clear the virus more effectively [28]. Also, according to other authors, an initial immune response launched against HCV in the genital mucosa, in cases of anal intercourse, could contribute to subsequent immunological events and subsequent viral clearance [31]. Repeated HCV exposures and reinfections among vulnerable populations could increase effective HCV-specific immunity. Spontaneous HCV clearance after HCV reinfection has been demonstrated in chimpanzees and humans. In such cases, rapid control of viral replication and greater likelihood of spontaneous resolution of secondary infection, suggestive of partial protective immunity, have been described [36].

According to our data the only variable independently associated with spontaneous HCV clearance was the CD4+ T lymphocyte count at the time of HCV infection. This reinforces the role of the CD4+ T lymphocyte count in HCV-specific immunity, as has been postulated in different studies [36]. In addition, several studies have now demonstrated that control of reinfection is likely associated with the magnitude of HCV-specific T-cell responses [37,38,39]. 

Limitations in our study should be acknowledged. First, we were not able to determine the polymorphisms in the interleukin-28 (IL-28) gene in our cohort due to the lack of blood samples. Second, the relatively small sample size limited our ability to conduct more complete analyses. Third, this investigation included patients from a single reference center. Thus, the observed findings may not be reflective of parameters of individuals in centers in other regions of Brazil and elsewhere.

The Brazilian Ministry of Health over the last 10 years has extended DAA treatment to everyone with an HCV diagnosis and has launched additional initiatives focused on HCV diagnosis and prevention. Our data suggest that HCV-preventive actions should be intensified, especially in PLWH and in MSM, subgroups of individuals with increased vulnerability. These actions should include programs and policies aimed at reducing sexual risk—such as regular testing for HCV and other infections—the promotion of safe sex practices, and coordination between treatment services and centers for sexually transmitted infection testing and counseling. 

These interventions could also help in modeling future morbidity, mortality, and costs related to HCV infection in this population and could make HCV elimination faster and more feasible in Brazil, as proposed by the World Health Organization to achieve global HCV elimination.

## 5. Conclusions

In HIV-infected individuals in Sao Paulo, Brazil, most cases of recent HCV transmission result from sexual exposure. In PLWH, and particularly in MSM, the individual’s CD4+ T lymphocyte count is a determinant of whether an acquired HCV infection will be prolonged or spontaneously clear.

## Figures and Tables

**Table 1 viruses-15-00314-t001:** Clinical characteristics of the 48 HIV-positive patients who recently acquired HCV infection.

Variables	Value (%)
Age at time of HIV diagnosis (years)	
Mean ± SD	31.2 ± 8.3
Median (min.; max.)	30.5 (19; 53)
Age at time of HCV diagnosis (years)	
Mean ± SD	46.7 ± 10.5
Median (min.; max.)	48 (23; 73)
Gender n (%)	
Female	8 (16.7)
Male	40 (83.3)
Time between HIV and HCV diagnosis (years)	
Mean ± SD	15.6 ± 7.4
Median (min.; max.)	17.5 (0.9; 27.8)
CD4+ T lymphocyte count at the time of HIV diagnosis	
Mean ± SD	364.7 ± 271.6
Median (min.; max.)	31.5 (9; 1199)
CD4+ T lymphocyte count at the time of HIV diagnosis, n (%)	
≥200	36 (75)
<200	12 (25)
Number of comorbidities and opportunistic infections before HCV diagnosis	
Mean ± SD	3.48 ± 1.82
Median (min.; max.)	3 (0; 8)
CD4+ T lymphocyte count at the time of HCV diagnosis	
Mean ± SD	753.2 ± 297.9
Median (min.; max)	729.5 (120; 1537)
CD4+ T lymphocyte count at the time of HCV diagnosis, n (%)	
≥200	46 (95.8)
<200	2 (4.2)
HIV viral load at the time of HCV diagnosis, n (%) **	
Undetectable	41 (85.4)
Detectable	7 (14.6)
AST at the time of HCV diagnosis	
Mean ± SD	2.98 ± 6.48
Median (min.; max)	0 (0; 32)
AST times upper limit of normal	
<3 times	37 (77.1)
3–5 times	4 (8.3)
>5 times	7 (14.6)
ALT at the time of HCV diagnosis	
Mean ± SD	4.4 ± 7.76
Median (min.; max)	1 (0; 32)
ALT times upper limit of normal	
<3 times	31 (64.6)
3–5 times	6 (12.5)
>5 times	11 (22.9)
History of chronic liver disease, n (%) *	
No	44 (93.6)
Yes	3 (6.4)
Spontaneous clearance of HCV, n (%)	
No	25 (52.1)
Yes	23 (47.9)

* Data were missing for one patient; ** Viral load was grouped as above or below assay lower limit of detection. The lower limit of detection varied according to method over the years between 20 and 40 copies/mL.

**Table 2 viruses-15-00314-t002:** Clinical and laboratory characteristics of 48 patients coinfected with HIV and HCV with and without spontaneous HCV clearance.

Variable	Spontaneous Clearance		OR	IC (95%)		*p*
	No	Yes		lower	upper	
Age at the time of HIV diagnosis (years)			1.013	0.945	1.086	0.725 **
Mean ± SD	30.8 ± 8,8	31.7 ± 7.9				
Median (min.; max)	28 (19; 53)	33 (19; 51)				
Age at the time of HCV diagnosis (years)			1.038	0.98	1.1	0.203 **
Mean ± SP	44.8 ± 10	48.7 ± 11				
Median (min.; max)	45(28; 73)	50 (23; 71)				
Gender N (%)						0.020 *
Female	1 (12.5)	7 (87.5)	10.5	1.18	93.7	
Male	24 (60)	16 (40)	1.00			
Time between HIV and HCV diagnosis (years)			1.064	0.979	1.155	0.167 £
Mean ± SD	14.1 ± 7.5	17.2 ± 7				
Median (min.; max)	14.6 (0.9; 25.8)	19.1 (1.9; 27.8)				
CD4+ T lymphocyte count at the time of HIV diagnosis			1. 000	0.997	1.002	0.628 £
Mean ± SD	379.1 ± 274.7	348.9 ± 273.4				
Median (min.; max)	326 (11; 1122)	317 (9; 1199)				
CD4+ T lymphocyte count at the time of HIV diagnosis, N (%)						0.404
≥200	20 (55.6)	16(44.4)	1. 00			
<200	5 (41.7)	7 (58.3)	1.75	0.47	6.57	
Number of comorbidities			1.052	0.768	1.441	0.809 £
Median ± SD	3.4 ± 1.63	3.57 ± 2.04				
Median (min.; max)	3 (1; 8)	3 (0; 8)				
CD4+ T lymphocyte count at the time of HCV diagnosis			1.004	1.001	1.007	0.006 £
Mean ± SD	629.8 ± 229.3	887.4 ± 310.1				
Median (min.; max)	620 (120; 1139)	834 (464; 1537)				
CD4+ T lymphocyte count at the time of HCV diagnosis, N (%)						0.490 *
≥200	23 (50)	23 (50)	1. 00			
<200	2 (100)	0 (0)	&			
HIV viral load at the time of HCV diagnosis, N (%)						0.419 *
Undetectable	20 (48.8)	21 (51.2)	1.00			
Detectable	5 (71.4)	2 (28.6)	0.38	0.07	2.19	
AST at the time of HCV diagnosis, N (%)			0.974	0.888	1.069	<0.001 £
Mean ± SD	3.48 ± 4.75	2.43 ± 8.03				
Median (min.; max)	2 (0; 18)	0 (0; 32)				
AST times above upper limit of normal N(%)						0.024 *#*
<3 times	16 (43.2)	21 (56.8)	3.28	0.56	19.15	
3–5 times	4 (100)	0 (0)	&			
>5 times	5 (71.4)	2 (28.6)	1.00			
ALT at the time of HCV diagnosis, N (%)			0.942	0.863	1.029	<0.001 £
Mean ± SD	5.88 ± 6.7	2.78 ± 8.62				
Median (min.; max)	3 (0; 23)	0 (0; 32)				
ALT times above upper limit of normal, N (%)						<0.001 #
<3 times	10 (32; 3)	21 (67.7)	9.45	1.71	52.1	
3–5 times	6 (100)	0 (0)	&			
>5 times	9 (81.8)	2 (18.2)	1.00			
History of chronic liver disease, N (%)						0.234 *
No	21 (47.7)	23 (52.3)	1.00			
Yes	3 (100)	0 (0)	&			

Chi-square test; * exact Fisher’s tests; # likelihood ratio; ** Student *t*-test; £ Mann –Whitney U-test; & it is not possible to estimate.

**Table 3 viruses-15-00314-t003:** Spontaneous HCV clearance in 23 patients coinfected with HIV and HCV: multiple logistic regression analysis.

Variable	OR	Confidence Interval		
		Inferior	Superior	*p*
Gender (female)	2.83	0.26	31.28	0.395
CD4+ T lymphocyte count at the time of HCV diagnosis	1.45	1.05	2.01	0.025
AST times above upper limit of normal	4.57	0.9	23.29	0.067
ALT times above upper limit of normal	0.27	0.07	1.05	0.050

Multiple logistic regression analysis (full model).

## Data Availability

Not applicable.

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
