# Peer review of "Factors Associated with Spontaneous Clearance of Recently Acquired Hepatitis C Virus among HIV-Positive Men in Brazil"

_viruses, 2023, doi:10.3390/v15020314_

Round 1

Reviewer 1 Report (Previous Reviewer 1)

The manuscript by Ferrufino et al reports retrospective studies of HCV infection in an HIV-infected cohort in Brazil. The manuscript has been improved by inclusion of the details of the RT-PCR and immunoassays used for analysis of HCV infection.

The manuscript will be further strengthened by description of the direct acting antivirals (DAA) used to treat acute and chronic HCV infection and description of the resulting sustained virological response (SVR).

Explanation of the differences between the SVR achieved by DAA treatment and the spontaneous clearance of HCV infection will add significantly to the manuscript by providing further understanding of these different outcomes.

The manuscript also requires thorough proofreading as it still contains many typos, punctuation and spacing errors particularly in the Tables.

1.       Line 16. Provide abbreviation for HIV on line 16 and remove the full name from line 17.

2.       Lines 26, 29, 102, 108, 136, Table 1 (4 instances), 168, 170, 171, Table 2 (4 instances), Table 3 (1 instance), 250, 251 and 276. In all instances revise to “CD4+ T lymphocyte count”.

3.       Line 30. Revise to “will spontaneously clear”.

4.       Lines 39-40. Revise to “has been the major route”.

5.       Line 57. Revise to “The proportion of patients who spontaneously clear HCV infection”.

6.       Line 58. Revise to “20-50%”.

7.       Line 59. Delete “viral”.

8.       Line 60, 234, 256. Provide full name of the IL28 C/C genotype on line 60 and use consistent abbreviation in all instances.

9.       Line 65. Introduce the direct DAA used for the treatment of acute and chronic HCV infection. Describe the mechanism of action of the DAA and the characteristics of the resulting SVR.

10.   Lines 86-87. Revise to “previous treatment for HCV”.

11.   Lines 94-95. Revise to “tested using the Abbott Real Time HCV genotype II assay”.

12.   Lines 98 and 146. Revise to “mL”.

13.   Line 109. Add a Section that provides details of the DAA compounds, their mechanism(s) of action and the definition of SVR with respect to the timing of DAA treatment and the resulting levels of HCV RNA.

14.   Line 103. Provide full names for ALT and AST. Revise to “[U/L]”.

15.   Line 104. Revise to “HCV diagnosis. The time between”.

16.   Line 144. Revise the heading to Table 1 to “of the 48 HIV positive patients who recently”.

17.   Table 1. Check spacing of: Mean +/- SD, n (%), (min; max), < 200, < 3, > 3, > 5,

18.   Table 1. The heading “N (%)” does not apply to all variables. Could it be replaced with “Value”?

19.   Table 1. The “n (%)” label is not needed for “AST/ALT at time of diagnosis”.

20.   Table 1. Revise to “ALT times upper limit of normal”.

21.   Table 1. Revise to “Spontaneous clearance of HCV, n (%)”.

22.   Table 1. Check spacing of numbers and +/- in Column 2.

23.   Lines 147-166. This Section is confusing, particularly the information on clinical outcomes of HCV infection. The Section could start with the statement “Spontaneous HCV clearance occurred in 23 individuals (47.9%).

24.   Lines 151-166. This Section must be revised in light of the previous comments about DAA and SVR and must explain the differences between the SVR achieved by DAA treatment and the spontaneous clearance of HCV infection.

25.   Line 155. Revise to “normal”.

26.   Line 157. Revise to “HCV PCR positive”.

27.   Line 167. Revise to “In univariate analysis, spontaneous HCV clearance”.

28.   Table 2.  Heading. Revise to “with and without spontaneous HCV clearance”.

29.   Table 2. Check spacing and spelling of: Mean +/- SD, < 200, n (%), < 3, > 5. Add “(min; max)” after Median.

30.   Table 3. Revise using footnotes to explain how the data was determined. For example, was the OR determined by bivariate logistic regression? The Inferior and Superior values are not explained. It needs to be clarified if the P value represents the results of the multivariate analysis.

31.   Table 3. Revise heading to “HIV and HCV: logistic regression analysis”.

32.   Line 182-83. Revise to “spontaneous clearance of HCV infection was identified”.

33.   Line 227, Revise to “3, 6, 12 and 24”.

34.   Line 229. Revise to “without DAA treatment”.

35.   Line 249. Revise to “spontaneous HCV clearance”.

36.   Line 261. Revise to “extended DAA treatment”.

Author Response

Answer- We thank Reviewer 1 for the careful evaluation of our manuscript.

We are grateful for the comments and suggestions.

Based on your comments, we revised the paper. A point-by-point response to your concerns is shown below. 

Please also find attached a new version of the manuscript.

We will be happy to send you any further information you may require.

The manuscript by Ferrufino et al reports retrospective studies of HCV infection in an HIV-infected cohort in Brazil. The manuscript has been improved by inclusion of the details of the RT-PCR and immunoassays used for analysis of HCV infection.

The manuscript will be further strengthened by description of the direct acting antivirals (DAA) used to treat acute and chronic HCV infection and description of the resulting sustained virological response (SVR).

Answer- We thank the Reviewer for this important observation. As requested, we have modified the text to address this point. We have added a new section (HCV treatment). Please check modifications on lines 111-118 and 171-172.

Explanation of the differences between the SVR achieved by DAA treatment and the spontaneous clearance of HCV infection will add significantly to the manuscript by providing further understanding of these different outcomes.

The manuscript also requires thorough proofreading as it still contains many typos, punctuation, and spacing errors particularly in the Tables.

  1. Line 16. Provide abbreviation for HIV on line 16 and remove the full name from line 17.

Answer- We have done as suggested. Please check modification on lines 16-17.

  1. Lines 26, 29, 102, 108, 136, Table 1 (4 instances), 168, 170, 171, Table 2 (4 instances), Table 3 (1 instance), 250, 251 and 276. In all instances revise to “CD4+ T lymphocyte count”.

Answer- We have done as suggested. Please check modifications on lines 26, 29, 102, 108, 145, Table 1 (4 instances), 178, 180, 181, Table 2 (4 instances), Table 3 (1 instance), 263, 264 and 288-289.

  1. Line 30. Revise to “will spontaneously clear”.

Answer- We have done as suggested. Please check modification on line 30.

  1. Lines 39-40. Revise to “has been the major route”.

Answer- We have done as suggested. Please check modification on lines 39.

  1. Line 57. Revise to “The proportion of patients who spontaneously clear HCV infection”.

Answer- We have done as suggested. Please check modification on line 56.

  1. Line 58. Revise to “20-50%”.

Answer- We have done as suggested. Please check modification on line 57.

  1. Line 59. Delete “viral”.

Answer- We have done as suggested. Please check modification on line 58.

  1. Line 60, 234, 256. Provide full name of the IL28 C/C genotype on line 60 and use consistent abbreviation in all instances.

Answer- We have done as suggested. Please check modifications on lines 59, 248 and 269.

  1. Line 65. Introduce the direct DAA used for the treatment of acute and chronic HCV infection. Describe the mechanism of action of the DAA and the characteristics of the resulting SVR.

Answer- We thank the Reviewer for this important observation. We have added a Section on HCV treatment. Please check lines 112-118 and lines 171-175.

Regarding information on the mechanisms of action of the DAAs, we respectfully disagree with the reviewer, and we decided not to include this type of information, because we believe this type of information and discussion is beyond the scope of the present manuscript.

  1. Lines 86-87. Revise to “previous treatment for HCV”.

Answer- We have done as suggested. Please check modifications on lines 86-87.

  1. Lines 94-95. Revise to “tested using the Abbott Real Time HCV genotype II assay”.

Answer- We have done as suggested. Please check modifications on lines 94-95.

  1. Lines 98 and 146. Revise to “mL”.

Answer- We have done as suggested. Please check modifications on lines 98 and 156.

  1. Line 109. Add a Section that provides details of the DAA compounds, their mechanism(s) of action and the definition of SVR with respect to the timing of DAA treatment and the resulting levels of HCV RNA.

Answer- We thank the Reviewer for this important observation. We have added a Section on HCV treatment. Please check lines 112-118 and lines 171-175.

  1. Line 103. Provide full names for ALT and AST. Revise to “[U/L]”.

Answer- We thank the Reviewer for this observation. We have included this information as requested. Please check line 103.

  1. Line 104. Revise to “HCV diagnosis. The time between”.

Answer- We have done as suggested. Please check modifications on lines 104-105.

  1. Line 144. Revise the heading to Table 1 to “of the 48 HIV positive patients who recently”.

Answer- We have done as suggested. Please check modifications on lines 153-154.

  1. Table 1. Check spacing of: Mean +/- SD, n (%), (min; max), < 200, < 3, > 3, > 5,

Answer- We thank the Reviewer for this important observation. We have done as suggested. Please check Table 1.

  1. Table 1. The heading “N (%)” does not apply to all variables. Could it be replaced with “Value”?

Answer- We thank the Reviewer for this observation. We have done as suggested. Please check Table 1.

  1. Table 1. The “n (%)” label is not needed for “AST/ALT at time of diagnosis”.

Answer- We thank the Reviewer for this observation. Please check Table 1.

  1. Table 1. Revise to “ALT times upper limit of normal”.

Answer- We thank the Reviewer for this observation. Please check modifications on Table 1.

  1. Table 1. Revise to “Spontaneous clearance of HCV, n (%)”.

Answer- We have done as suggested. Please check modifications on Table 1.

  1. Table 1. Check spacing of numbers and +/- in Column 2.

Answer- We have done as suggested. Please check modifications on Table 1, Column 2.

  1. Lines 147-166. This Section is confusing, particularly the information on clinical outcomes of HCV infection. The Section could start with the statement “Spontaneous HCV clearance occurred in 23 individuals (47.9%).

Answer- We thank the Reviewer for this observation. Nevertheless, we believe that the information on this section (Results) is clearer after we revised the manuscript according to the suggestions made by the Reviewers.

  1. Lines 151-166. This Section must be revised in light of the previous comments about DAA and SVR and must explain the differences between the SVR achieved by DAA treatment and the spontaneous clearance of HCV infection.

Answer- We thank the Reviewer for this observation. As requested, we have added information on HCV treatment. Please check lines 111-117 and 170-176.

  1. Line 155. Revise to “normal”.

Answer- I apologize, but I was not able to find “normal” on Line 155, or any word which could be revised to“normal”.

  1. Line 157. Revise to “HCV PCR positive”.

Answer- We have done as suggested. Please check modifications on line 160.

  1. Line 167. Revise to “In univariate analysis, spontaneous HCV clearance”.

Answer- We have done as suggested. Please check modifications on line 177.

  1. Table 2.  Heading. Revise to “with and without spontaneous HCV clearance”.

Answer- We have done as suggested. Please check modifications on Table 2.

  1. Table 2. Check spacing and spelling of: Mean +/- SD, < 200, n (%), < 3, > 5. Add “(min; max)” after Median.

Answer- We have done as suggested. Please check modifications on Table 2.

  1. Table 3. Revise using footnotes to explain how the data was determined. For example, was the OR determined by bivariate logistic regression? The Inferior and Superior values are not explained. It needs to be clarified if the P value represents the results of the multivariate analysis.

Answer- We thank the Reviewer for this observation. We have added a footnote in order to clarify how data was determined.

Details on Statistics are described in the Statistic Section. Please refer to lines 119-133 (2.5. Statistic Section).

  1. Table 3. Revise heading to “HIV and HCV: logistic regression analysis”.

Answer- Please check modifications on Table 3.

  1. Line 182-83. Revise to “spontaneous clearance of HCV infection was identified”.

Answer- We have done as suggested. Please check modifications on lines 195-196.

  1. Line 227, Revise to “3, 6, 12 and 24”.

Answer- We have done as suggested. Please check modifications on lines 240-241.

  1. Line 229. Revise to “without DAA treatment”.

Answer- We have done as suggested. Please check modifications on lines 242-243.

  1. Line 249. Revise to “spontaneous HCV clearance”.

Answer- We have done as suggested. Please check modifications on lines 262-263.

  1. Line 261. Revise to “extended DAA treatment”.

Answer- We have done as suggested. Please check modifications on line 274.

Reviewer 2 Report (Previous Reviewer 3)

This retrospective cross-sectional study describes the clinical and epidemiological aspects of recently acquired HCV infection and the frequency of its spontaneous clearance in a sample of PLWH in Brazil.

The authors revised the manuscript and have responded to the comments of the three reviewer’s point-to-point. However, there is a concern related with the definition of recently acquired hepatitis C. In this study the diagnosis of recently acquired hepatitis C was based “as being anti-HCV antibody positive or anti-HCV antibody and HCV-RNA positive beginning no earlier than January 2015 with negative anti-HCV tests in the previous five years. The seroconversion date was defined as the first positive anti-HCV test”. The criteria for definition of recently acquired hepatitis C in this study is far from the criteria for definition generally considered, with particularly implications in the cases of those PLWH who had no spontaneous HCV clearance. First, we assume that this tests were done in the context of high-risk population (as it’s the case of PLWH) as routine HCV screening. Second, recently acquired HCV infection is defined by the presence of anti-HCV antibodies, HCV RNA and/or HCV core antigen that were not detectable in previous samples up to 12 months. If such samples are unavailable, the diagnosis of recently acquired hepatitis C is based on the presence of HCV RNA or HCV core antigen, in the presence or absence of anti-HCV antibodies associated with a 3-fold or greater rise in ALT levels above baseline in an individual who had a risk behaviour in the preceding 6 months. Third, in individuals living with HIV, early chronic HCV infection can be defined as an estimated duration of infections < 12 months and a lack of a 2-log reduction of HCV RNA levels 4 weeks after initial presentation with recently acquired hepatitis C. So the definition of recently acquired HCV infection endorsed by the authors do not match with the consensual definition.

ABSTRACT

Line 16 …with human immunodeficiency virus (PLWH) cohort

Line 20 …Among 3, 143 PLWH, 362 (11.5%) were co-infected…

Line 29 …CD4+ lymphocyte T-cells count is a determinant…

INTRODUCTION

Line 46 …higher from HCV-human immunodeficiency virus (HIV) co-infected mothers (5)

Line 51 …transmitted disease and HIV infection (9,10)

Line 56 …infection in the PLWH

Line 58 …to range from 20-50% to 5-20% among mono…

MATERIALS AND METHODS

Lines 76 and 77 …HCV infection was diagnosed by the presence of anti-HCV antibodies (note: recently acquired or chronic infection is based on the presence of HCV RNA)

Line 85 …in at least two consecutive serum samples,…

Line 108 …CD4+ lymphocyte T-cells count were extracted from data during the six months prior to…

RESULTS

Line 127 …being HCV-infected based on anti-HCV antibody positivity (note: those anti-HCV antibodies positive, together with the presence of HCV RNA are those with recently acquired or chronic hepatitis C infection)

Line 131 …median age of 31 years (23-39)

Line 141 …to HBV core antigen, while 23 were positive…

Line 144 …Table 1. Clinical and laboratory characteristics of 48 PLWH who recently acquired HCV infection

Line 148 …related to their recently acquired HCV infection…

Line 149 …in the previous 6-12 months)…

Line 152 …Among them, five were treated…

Line 153 …a sustained virologic response (SVR)

Line 159 …subjects 21 received HCV treatment…

Lines 165 and 166 …longer than six months. A SVR was observed in all 21 patients

Line 169 …AST elevation levels six months prior…

Line 173 and 174 …Table 2. Clinical and laboratory characteristics of 48 patients HCV-HIV co-infected, who had or not spontaneous HCV clearance

Line 177 …Table 3. Spontaneous HCV clearance in 23 patients co-infected…

Line 188 …blood transfusion and IVDU were the main…

Line 189 …and among HCV-HIV…

Line 202 …changes in the safety of blood transfusion…

Line 215 …and only seven reported mild to moderate symptoms related to their recently acquired HCV infection

Line 216 …of recently acquired HCV infection…

Lines 219-222 – Please correct, because the find of 47,9% is lower than 49%.

Lines 250, 251 and 276 – Please correct to CD4+ lymphocyte T-cells count

Author Response

Answer- We thank Reviewer 2 for the careful evaluation of our manuscript.

We are grateful for the comments and suggestions.

Based on your comments, we revised the paper. A point-by-point response to your concerns is shown below. 

Please also find attached a new version of the manuscript.

We will be happy to send you any further information you may require.

This retrospective cross-sectional study describes the clinical and epidemiological aspects of recently acquired HCV infection and the frequency of its spontaneous clearance in a sample of PLWH in Brazil.

The authors revised the manuscript and have responded to the comments of the three reviewer’s point-to-point. However, there is a concern related with the definition of recently acquired hepatitis C. In this study the diagnosis of recently acquired hepatitis C was based “as being anti-HCV antibody positive or anti-HCV antibody and HCV-RNA positive beginning no earlier than January 2015 with negative anti-HCV tests in the previous five years. The seroconversion date was defined as the first positive anti-HCV test”. The criteria for definition of recently acquired hepatitis C in this study is far from the criteria for definition generally considered, with particularly implications in the cases of those PLWH who had no spontaneous HCV clearance. First, we assume that this tests were done in the context of high-risk population (as it’s the case of PLWH) as routine HCV screening. Second, recently acquired HCV infection is defined by the presence of anti-HCV antibodies, HCV RNA and/or HCV core antigen that were not detectable in previous samples up to 12 months. If such samples are unavailable, the diagnosis of recently acquired hepatitis C is based on the presence of HCV RNA or HCV core antigen, in the presence or absence of anti-HCV antibodies associated with a 3-fold or greater rise in ALT levels above baseline in an individual who had a risk behaviour in the preceding 6 months. Third, in individuals living with HIV, early chronic HCV infection can be defined as an estimated duration of infections < 12 months and a lack of a 2-log reduction of HCV RNA levels 4 weeks after initial presentation with recently acquired hepatitis C. So, the definition of recently acquired HCV infection endorsed by the authors do not match with the consensual definition.

 Answer- We thank the Reviewer for this observation. It gave us an opportunity to clarify certain aspects related to the criteria for definition of recently acquired hepatitis C in our study.

In Brazil, in the past, blood transfusion and illicit IVDU were the main risk factors associated with HCV acquisition in the general population and among HIV-HCV co-infected individuals (18). Recently however, outbreaks of recently acquired HCV infections in men who have sex with men (MSM), particularly in those living with HIV (PLWH), have been observed globally. Most of these outbreaks have been described in Europe, Australia, and USA (11-15).  However, in Brazil and other countries in Latin America, there is a paucity of data on the mode of transmission of HCV infection in the PLWH population.  The objective of the present study was to describe epidemiological aspects of recently acquired HCV infection and the frequency of its spontaneous clearance in a cohort of individuals who acquired HCV infection no earlier than January 2015.

All included individuals were diagnosed with HCV infection by detection of anti-HCV antibody or anti-HCV antibody and HCV-RNA with a documented anti-HCV antibody negative test in the previous 6 to 12 months. So, our definition of recently acquired hepatitis C combines the consensual definition of recently acquired hepatitis C with the fact that all infections occurred in recent years: no earlier than January 2015.

Line 16 …with human immunodeficiency virus (PLWH) cohort

Answer- We have done as suggested. Please check modification on line 16.

Line 20 …Among 3, 143 PLWH, 362 (11.5%) were co-infected…

Answer- We have done as suggested. Please check modification on line 22.

Line 29 …CD4+ lymphocyte T-cells count is a determinant…

 Answer- We have done as suggested. Please check modification on line 29.

INTRODUCTION

Line 46 …higher from HCV-human immunodeficiency virus (HIV) co-infected mothers (5)

Answer- We have done as suggested. Please check modification on lines 45-46

Line 51 …transmitted disease and HIV infection (9,10)

Answer- We have done as suggested. Please check modification on lines 50-51.

Line 56 …infection in the PLWH

Answer- We have done as suggested. Please check modification on line 55.

Line 58 …to range from 20-50% to 5-20% among mono…

Answer- We have done as suggested. Please check modification on line 57.

MATERIALS AND METHODS

Lines 76 and 77 …HCV infection was diagnosed by the presence of anti-HCV antibodies (note: recently acquired or chronic infection is based on the presence of HCV RNA)

Answer- We thank the Reviewer for this observation. It gave us an opportunity to clarify certain aspects related to the criteria for definition of recently acquired hepatitis C in our study. Please refer to Section 2.1, Study Design and Population, lines 69-82.

Line 85 …in at least two consecutive serum samples, …

Answer- We have done as suggested. Please check modification on line 85.

Line 108 …CD4+ lymphocyte T-cells count were extracted from data during the six months prior to…

 Answer- We have done as suggested. Please check modification on line 109.

RESULTS

Line 127 …being HCV-infected based on anti-HCV antibody positivity (note: those anti-HCV antibodies positive, together with the presence of HCV RNA are those with recently acquired or chronic hepatitis C infection)

Answer- We thank the Reviewer for this observation. It gave us an opportunity to clarify certain aspects related to the criteria for definition of recently acquired hepatitis C in our study. Please refer to Section 2.1, Study Design and Population, lines 69-82.

Line 131 …median age of 31 years (23-39)

Answer- We have done as suggested. Please check modification on line 140.

Line 141 …to HBV core antigen, while 23 were positive…

Answer- We have done as suggested. Please check modification on line 149.

Line 144 …Table 1. Clinical and laboratory characteristics of 48 PLWH who recently acquired HCV infection

Answer- We have done as suggested. Please check modification on line 153.

Line 148 …related to their recently acquired HCV infection…

Answer- We have done as suggested. Please check modification on line 159.

Line 149 …in the previous 6-12 months)…

Answer- We have done as suggested. Please check modification on line 160.

Line 152 …Among them, five were treated…

Answer- We have done as suggested. Please check modification on line 163.

Line 153 …a sustained virologic response (SVR)

Answer- We have done as suggested. Please check modification on line 164.

Line 159 …subjects 21 received HCV treatment…

Answer- We have done as suggested. Please check modification on line 170.

Lines 165 and 166 …longer than six months. A SVR was observed in all 21 patients

Answer- We have done as suggested. Please check modification on line 175.

Line 169 …AST elevation levels six months prior…

Answer- We have done as suggested. Please check modification on line 179.

Line 173 and 174 …Table 2. Clinical and laboratory characteristics of 48 patients HCV-HIV co-infected, who had or not spontaneous HCV clearance

Answer- We have done as suggested. Please check modification on lines 184-185.

Line 177 …Table 3. Spontaneous HCV clearance in 23 patients co-infected…

Answer- We have done as suggested. Please check modification on line 188.

Line 188 …blood transfusion and IVDU were the main…

Answer- We have done as suggested. Please check modification on line 201.

Line 189 …and among HCV-HIV…

Answer- We have done as suggested. Please check modification on line 202.

Line 202 …changes in the safety of blood transfusion…

Answer- We have done as suggested. Please check modification on line 215.

Line 215 …and only seven reported mild to moderate symptoms related to their recently acquired HCV infection.

Answer- We have done as suggested. Please check modification on lines 228-229.

Line 216 …of recently acquired HCV infection…

Answer- We have done as suggested. Please check modification on lines 229-230.

Lines 219-222 – Please correct, because the find of 47,9% is lower than 49%.

Answer- We respectfully disagree with the reviewer regarding this observation. According to previous studies, spontaneous HCV clearance has been observed from 11% to 49%, among different population of recently acquired HCV infection. Our intention is to compare our results with these previous results. In fact, our results may be compared to the highest results observed in literature, as mentioned in our manuscript.

Lines 250, 251 and 276 – Please correct to CD4+ lymphocyte T-cells count

Answer- We thank the reviewer for this observation. Please check modifications on lines 263,264,288.

Round 2

Reviewer 2 Report (Previous Reviewer 3)

The authors revised the manuscript, and replied point-by-point to the comments and suggestions. Minor suggestions remained:

Lines 77 and 83 - Please correct to 6-12 months as it is in line 161

Line 105 - Please correct to alanine transaminase (ALT), and aspartate transaminase (AST)

Line 119 - Please correct to "...and six months later".

Line 165 - Please correct to "...reached a SVR".

Lines 239 and 240 - Please correct to "...within three, six, 12 and 24 months...".

Author Response

(Reviewer 2)-R3

Answer- We thank Reviewer 2 for the careful evaluation of our manuscript.

We are grateful for the comments and suggestions.

Based on your comments, a point-by-point response to your concerns is shown below. 

Please also find attached a new version of the manuscript.

We will be happy to send you any further information you may require.

The authors revised the manuscript and replied point-by-point to the comments and suggestions. Minor suggestions remained:

Lines 77 and 83 - Please correct to 6-12 months as it is in line 161

Answer- We have done as suggested. Please check modification on lines 76 and 82.

Line 105 - Please correct to alanine transaminase (ALT), and aspartate transaminase (AST)

Answer- We have done as suggested. Please check modification on line 104.

Line 119 - Please correct to "...and six months later".

Answer- We have done as suggested. Please check modification on line 119.

Line 165 - Please correct to "...reached a SVR".

Answer- We have done as suggested. Please check modification on line 165.

Lines 239 and 240 - Please correct to "...within three, six, 12 and 24 months...".

Answer- We have done as suggested. Please check modification on lines 241 and 242.

This manuscript is a resubmission of an earlier submission. The following is a list of the peer review reports and author responses from that submission.

Round 1

Reviewer 1 Report

The manuscript by Ferrufino and colleagues reports retrospective studies of hepatitis C virus (HCV) infection in a human immunodeficiency virus (HIV)-infected cohort in Brazil. Interestingly, the authors found that HCV is transmitted primarily by sexual contact between men who had sex with men (MSM) and that the individual’s CD4+ lymphocyte count was a key determinant in the outcome of recently acquired HCV infection. The manuscript would be strengthened by including details of the RT-PCR, genotyping and immunoassays used for analysis of HCV infection. It’s important to clarify if HCV infection was diagnosed by detection of HCV RNA or anti-HCV antibodies and which assays were used to define seroconversion and spontaneous clearance of HCV infection.

The manuscript also requires thorough proof reading as it contains many typos, punctuation and spacing errors and instances where the text, particularly in the Tables, should be presented not in capitals but in lower case.

1.       Lines 4-5. Use consistent spacing throughout.

2.       Lines 6-11. Insert full stops after Brazil and USA.

3.       Lines 12-13. Remove “Street name:”, “CODE” and “Phone Number”. Insert a space before, and a full stop after, “Brazil”.

4.       Line 15. Provide the full name for the abbreviation HCV.

5.       Line 16. Provide the full name for the abbreviations PLWH and HIV.

6.       Lines 17-18. Revise to: and identified individuals who acquired HCV infection between…”

7.       Line 19. Insert “HCV” before “infection”.

8.       Line 20. Were the individuals “infected with HCV” identified by RT-PCR?

9.       Line 21. Subjects who “became HCV-positive” was this determined in all cases by RT-PCR? If RT-PCR and immunoassays were both used then say so.

10.   Line 22. Was the “spontaneous HCV clearance” determined in all cases by RT-PCR?

11.   Line 27. Provide the full name for the abbreviation IDU or IVDU. IVDU is used later in the manuscript.

12.   Lines 28, 78, 105, 133, 212, 213, 235, Table 1 (4 places), Table 2 (4 places), Table 3. Revise to “CD4+ lymphocyte”.

13.   Line 37. Include details of the different rates of HCV transmission through the different routes.

14.   Line 39. Provide the full name for the abbreviation HIV.

15.   Lines 43-44. Revise to: “paucity of data on the mode of transmission of HCV infection in the PLWH population.”

16.   Line 51. Check abbreviation “PLWHIV”.

17.   Line 55. Revise to “January 2015”.

18.   Lines 57-62. “Study Design and Population”. Include details of the RT-PCR, genotyping and immunoassays used for analysis of HCV infection. Include details of the “HIV viral load test” presented in Table 1. What is the sensitivity of the HIV viral load test? Clarify “HCV status” by defining “seroconversion” and “spontaneous HCV clearance”.

19.   Lines 64 and 65. Provide details of the “anti-HCV test”.

20.   Line 67. Revise to: “anti-HCV test”.

21.   Lines 70, 83 and 90. Revise to: “spontaneous clearance of HCV”.

22.   Line 75. Check abbreviation “HIC”.

23.   Line 98. Define how HCV infection was diagnosed.

24.   Line 106. Provide a reference to HCV genotypes.

25.   Line 109. Revise to: “anti-HBV surface antibodies”.

26.   Table 1. Requires careful proof reading. There are many problems with spacing and capitalization.  Is “n (%)” needed for each variable? It is given as the heading to column 2.

27.   Line 119. Delete “virus”.

28.   Line 129. Check abbreviation “HVC”.

29.   Table 2. Requires careful proof reading. There are many problems with spacing, capitalization. Columns do not line up. The footnotes are non-standard and confusing. Heading should read “spontaneous”.

30.   Table 3. Heading: Insert space before “Multiple”. Check column 1 as the labels for each variable are confusing.

31.   Line 145. Revise to: “high rate of spontaneous”.

32.   Line 179. Revise to: “acute HCV infection”.

33.   Line 182. Revise to: “of 48 individuals who recently acquired HCV infection”.

34.   Line 228. Define “STI”.

35.   Line 239. Check spacing.

36.   Line 249. Define “RL”.

37.   References. Proof read and check spacing. There are many instances where spaces appear after the year of publication.

38.   Line 276. Close space.

39.   Line 310. Move line up.

Reviewer 2 Report

Ferrufino and co-authors aimed to identify factors associated with HCV spontaneous clearance, defined as HCV RNA becoming undetectable without antiviral treatment within one year of first serological diagnosis, in treated patients living with HIV (PLWH). Out of 3,143 PLWH, 362 (11.5%) were anti-HCV positive and 48 (13.2%) fulfilled the study inclusion criteria. Spontaneous viral clearance was observed in 23 (47.9%) of these 48 HCV-infected patients. CD4 cell count at the time of first HCV diagnosis was the only factor reported positively associated with HCV clearance. Overall, these data from a small size cohort, as acknowledged by the authors, are quite limited.

Specific comments:

- Define PLWH abbreviation.

- The conclusion in the abstract that sexual transmission of HCV has replaced IDU as the most common mode of HCV acquisition in HIV-infected individuals may seem somewhat overstated as only 48 of 362 HCV and HIV dual infected patients were studied. At least, it should be rephrased, possibly limited to the types of patients studied.

- The patients studied were a very heterogeneous group with respect to the course of HCV infection, with the time interval between the last negative HCV serology and the first positive serology ranging from 13 months to 16 years. As spontaneous clearance rate is known to increase with time, as noted by the authors in the Discussion, this time interval parameter could have been considered.

- CD4 cell counts were available only at the time of HCV first diagnosis but it would be interesting to know how it evolved during follow-up. The lack of data on the specific immune response to anti-HCV is also an important limitation.

- Line 41 in Table 1: AST should be replaced by ALT.

- Tables 1 and 2 are quite redundant and could be merged in a single table.

- Table 3 is not called in the text.

Reviewer 3 Report

This article describes the factors associated with spontaneous clearance of recently acquired hepatitis C virus among HIV-positive MSM in Brazil. Characterizing spontaneous viral clearance of HCV infection among PWID and HIV + MSM is important for assessing the burden of disease and treatment strategies in these populations.

I have only a few comments:

INTRODUCTION

Please refer data reported in the literature on spontaneous viral clearance of acute HCV infection in people without HIV and in people with HIV. The same in PWID and MSM HIV+

PATIENT INCLUSION CRITERIA

Please define the criteria for acute HCV infection

RESULTS

Please refer which is the criteria to treat with DAAs acute HCV infection

MINOR REVISION

Line 57 …screening every six to 12 months

Line 65 …in the previous five years…

Line 107 …seven (31.8%) and two (9%), respectively

Line 110 …syphilis during the six months prior…

Line 114 …only seven…

Line 129 …had at least two HCV-PCR…

Line 189 …within three, six, 12 and 24 months